# Global distribution maps of the leishmaniases

David M Pigott[1]*, Samir Bhatt[1], Nick Golding[1], Kirsten A Duda[1], Katherine E Battle[1], Oliver J Brady[1], Jane P Messina[1], Yves Balard[2], Patrick Bastien[2,3], Francine Pratlong[2,3], John S Brownstein[4,5], Clark C Freifeld[5,6], Sumiko R Mekaru[5], Peter W Gething[1], Dylan B George[7], Monica F Myers[1], Richard Reithinger[8], Simon I Hay[1,7]

[1]Spatial Ecology and Epidemiology Group, Department of Zoology, University of Oxford, Oxford, United Kingdom; [2]Laboratoire de Parasitologie–Mycologie, UFR Médecine, Université Montpellier 1 and UMR 'MiVEGEC', CNRS 5290/IRD 224, Montpellier, France; [3]Departement de Parasitologie–Mycologie, CHRU de Montpellier, Centre National de Référence des Leishmanioses, Montpellier, France; [4]Department of Pediatrics, Harvard Medical School, Boston, United States; [5]Children's Hospital Informatics Program, Boston Children's Hospital, Boston, United States; [6]Department of Biomedical Engineering, Boston University, Boston, United States; [7]Fogarty International Center, National Institutes of Health, Bethesda, United States; [8]Global Health Group, RTI International, Washington DC, United States

**Abstract** The leishmaniases are vector-borne diseases that have a broad global distribution throughout much of the Americas, Africa, and Asia. Despite representing a significant public health burden, our understanding of the global distribution of the leishmaniases remains vague, reliant upon expert opinion and limited to poor spatial resolution. A global assessment of the consensus of evidence for leishmaniasis was performed at a sub-national level by aggregating information from a variety of sources. A database of records of cutaneous and visceral leishmaniasis occurrence was compiled from published literature, online reports, strain archives, and GenBank accessions. These, with a suite of biologically relevant environmental covariates, were used in a boosted regression tree modelling framework to generate global environmental risk maps for the leishmaniases. These high-resolution evidence-based maps can help direct future surveillance activities, identify areas to target for disease control and inform future burden estimation efforts.

*For correspondence: david. pigott@zoo.ox.ac.uk

Competing interests: The authors declare that no competing interests exist.

## Introduction

The leishmaniases are a group of protozoan diseases transmitted to humans and other mammals by phlebotomine sandflies (*Murray et al., 2005*; *WHO, 2010*). Considered as one of the neglected tropical diseases (NTD) (*WHO, 2009*), the leishmaniases can be caused by around 20 *Leishmania* species and include a complex life cycle involving multiple arthropod vectors and mammalian reservoir species (*Ashford, 1996*; *Ready, 2013*). Sandflies belonging to either *Phlebotomus* spp. (Old World) or *Lutzomyia* spp. (New World) are the primary vectors; domestic dogs, rodents, sloths, and opossums are amongst a long list of mammals that are either incriminated or suspected reservoir hosts. Non-vector transmission (e.g., by accidental laboratory infection, blood transfusion, or organ transplantation) is possible, but rare (*Cardo, 2006*). Transmission of the leishmaniases can be either anthroponotic or zoonotic. The leishmaniases rank as the leading NTD in terms of mortality and morbidity with an estimated 50,000 deaths in 2010 (*Lozano et al., 2012*) and 3.3 million disability adjusted life years (*Murray et al., 2012*).

**eLife digest** Each year 1–2 million people are diagnosed with a tropical disease called leishmaniasis, which is caused by single-celled parasites. People are infected when they are bitten by sandflies carrying the parasite, and often develop skin lesions around the bite site. Though mild cases may recover on their own or with treatment, sometimes the parasites multiply and spread elsewhere causing further skin lesions and facial disfigurement. Furthermore, the parasites can also infect internal organs such as the spleen and the liver, which without treatment can be fatal.

The parasites that cause leishmaniasis are found in 88 countries around the world, mainly in South and Central America, Africa, Asia, and southern Europe. However, over 90% of potentially fatal infections occur in just six countries: Brazil, Ethiopia, Sudan, South Sudan, India, and Bangladesh. Although a few studies have looked at the distribution of leishmaniasis within different countries, we still do not have a complete picture of the distribution of the disease on a global scale.

To address this, Pigott et al. set out to create detailed maps of the distribution of leishmaniasis and the factors that promote its spread. Similar techniques had been previously used to map dengue fever, another tropical disease. Computer modelling was used to generate the maps based on data about the environment at the locations of known cases of leishmaniasis. This information was then used to infer the likelihood of leishmaniasis being present at other locations across the globe.

Based on their maps, Pigott et al. estimate that about 1.7 billion people, or one quarter of the world's population, live in areas where they are at potential risk of leishmaniasis. People living in built-up areas outside of cities are at the greatest risk, likely because some sandfly species prefer to live near dwellings, but other social and economic factors also contribute to the risk of catching this disease.

The factors driving the transmission of leishmaniasis differed in the Old World (Europe, Africa and Asia) and the New World (the Americas): built-up areas were more likely to be at risk in the Old World, while temperature and rainfall were bigger factors affecting risk in the New World. It is hoped that the maps created by Pigott et al. will help inform future estimates of the burden of leishmaniasis and target surveillance and disease control efforts more effectively to combat this tropical disease.

---

Symptoms of *Leishmania* infection can take many different and diverse forms (*Banuls et al., 2011*), the two main outcomes being cutaneous leishmaniasis (CL) and visceral leishmaniasis (VL). Cutaneous leishmaniasis typically presents as cutaneous nodules or lesions at the site of the sandfly bite (localised cutaneous leishmaniasis). In some cases, parasites disseminate through the skin and present as multiple non-ulcerative nodules (diffuse cutaneous leishmaniasis, DCL) or propagate through the lymphatic system resulting in nasobronchial and buccal mucosal tissue destruction (mucosal leishmaniasis, ML) (*Reithinger et al., 2007*; *Dedet and Pratlong, 2009*). Localised CL may resolve spontaneously and usually responds well to treatment; management of DCL and ML cases is more difficult and cases may take considerably longer to resolve, if at all. Visceral leishmaniasis generally affects the spleen, liver, or other lymphoid tissues, and, if left untreated, is fatal; a fraction of successfully treated VL cases may result in maculopapular or nodular rashes (post-kala-azar dermal leishmaniasis) (*Murray et al., 2005*; *Dedet and Pratlong, 2009*). While the *Leishmania* species determines which of the main two forms of the leishmaniases will result from infection, establishment, progression, and severity of infection as well as treatment regimen and outcome is dependent on a range of other factors, including parasite strain, characteristics of sandfly saliva, parasite infection with *Leishmania* RNA virus, host genetics, and immunosuppression, particularly due to HIV co-infection (*Reithinger et al., 2007*; *Ives et al., 2011*; *Novais et al., 2013*).

Species distribution models provide a robust means of mapping these diseases at a global level. These models define a set of conditions, from a selection of environmental covariates, which best categorise known occurrences. Through this categorisation, areas of unknown pathogen presence can be identified and thus a global evaluation of environmental suitability for presence can be made.

A variety of factors can influence the distribution of an organism, including an array of environmental and other abiotic characteristics as well as biotic factors (*Peterson, 2008*). Whilst many areas may be environmentally suitable for a given species, other factors may prevent the species from being present in all of these locations. This distinction is often referred to as the difference between the fundamental and the realised niche of the species, the former describing a potential distribution based upon specific features of the environment whilst the latter indicates the distribution we observe. Such a framework can be applied just as successfully in the context of pathogens and their vectors as with macroorganisms (*Peterson et al., 2011*) and has already been applied to the mapping of malaria vectors (*Sinka et al., 2010*, *2010*, *2011*) and dengue (*Bhatt et al., 2013*). The relationships between the leishmaniases and environmental and socioeconomic factors known to influence their distribution at a global scale has not previously been considered in a comprehensive and quantitative manner (*Hay et al., 2013*). This study uses these modelling techniques in order to define the first evidence-based global environmental risk maps of the leishmaniases.

## Results

### Evidence of leishmaniasis

For each province or state across the globe (classed as Admin 1 by the Food and Agriculture Organization's Global Administrative Unit Layers (*FAO, 2008*), totalling some 3450) evidence was collected regarding CL and VL presence or absence. An assessment of the consensus of this evidence ranging from comprehensive agreement on disease presence (+100%) to consensus of disease absence (−100%) was made. *Figures 1A–4A* present these evidence consensus maps, with full reasoning for each administrative unit's score outlined in the associated data set (Dryad data set doi: 10.5061/dryad.05f5h). For Brazil, it was possible to perform this analysis at the district level (classed as Admin 2) totalling some 5510 units. In total, 950 Admin 1 units from 84 countries reported a consensus on CL presence greater than indeterminate (a score of 0), with 310 Admin 1 units from 42 countries reporting a complete consensus on the presence of CL. In Brazil, 2469 Admin 2 regions recorded CL cases over the period of investigation. Consensus on the presence of VL (score greater than 0) was reported in 793 Admin 1 units from 77 countries, with 88 Admin 1 units from 32 countries reporting complete consensus on VL. In Brazil, 1320 Admin 2 units recorded VL cases.

Of the 10 countries (Afghanistan, Colombia, Brazil, Algeria, Peru, Costa Rica, Iran, Syria, Ethiopia, and Sudan) that contribute 75% of the global estimated CL incidence (*Alvar et al., 2012*), only Algeria did not have regions of complete evidence consensus on presence due to incomplete and non-contemporary case data. Similarly, of the six countries (Brazil, Ethiopia, Sudan, South Sudan, India, and Bangladesh) that report 90% of all VL cases (*Alvar et al., 2012*), all six had regions of complete consensus on VL.

*Figures 1A–4A* also show the spatial distribution of occurrence data, defined as one or more reports of leishmaniasis in a given calendar year, collated from a variety of sources. Overall, there is a relatively broad geographic spread and good correspondence with the evidence consensus maps for each disease. Tunisia, Morocco and Brazil report the highest number of unique CL occurrences in any given year, whilst India reported the largest proportion of the VL occurrence data.

*Table 1* reports the sources and types of data within the occurrence database. Whilst the majority of occurrence records contain accurate point data (62%), the remainder were recorded at a provincial or district level. Occurrence records for the two diseases were relatively similar in number with a total of 6426 records for CL and 6137 for VL.

### Modelled distribution of the leishmaniases

*Figures 1B–4B* show the global predicted environmental risk maps for CL and VL. *Table 2* identifies the top five predictor variables in each of the four modelled regions (since CL and VL were modelled separately in the Old World and New World) as measured by average contribution to the boosted regression trees (BRT) submodels. Peri-urban and urban land cover is an important predictor of the distribution of CL in the Old World and of VL globally. Abiotic factors such as land surface temperature (LST) were better predictors of CL than of VL. In total, LST variables (annual minimum, maximum and mean) explain 21.99% of CL distribution in the Old World and 43.65% of CL distribution in the New World (with maximum LST having the highest relative contribution). Abiotic factors combined (including LST, normalised difference vegetation index (NDVI) and precipitation) accounted for 29.02% and

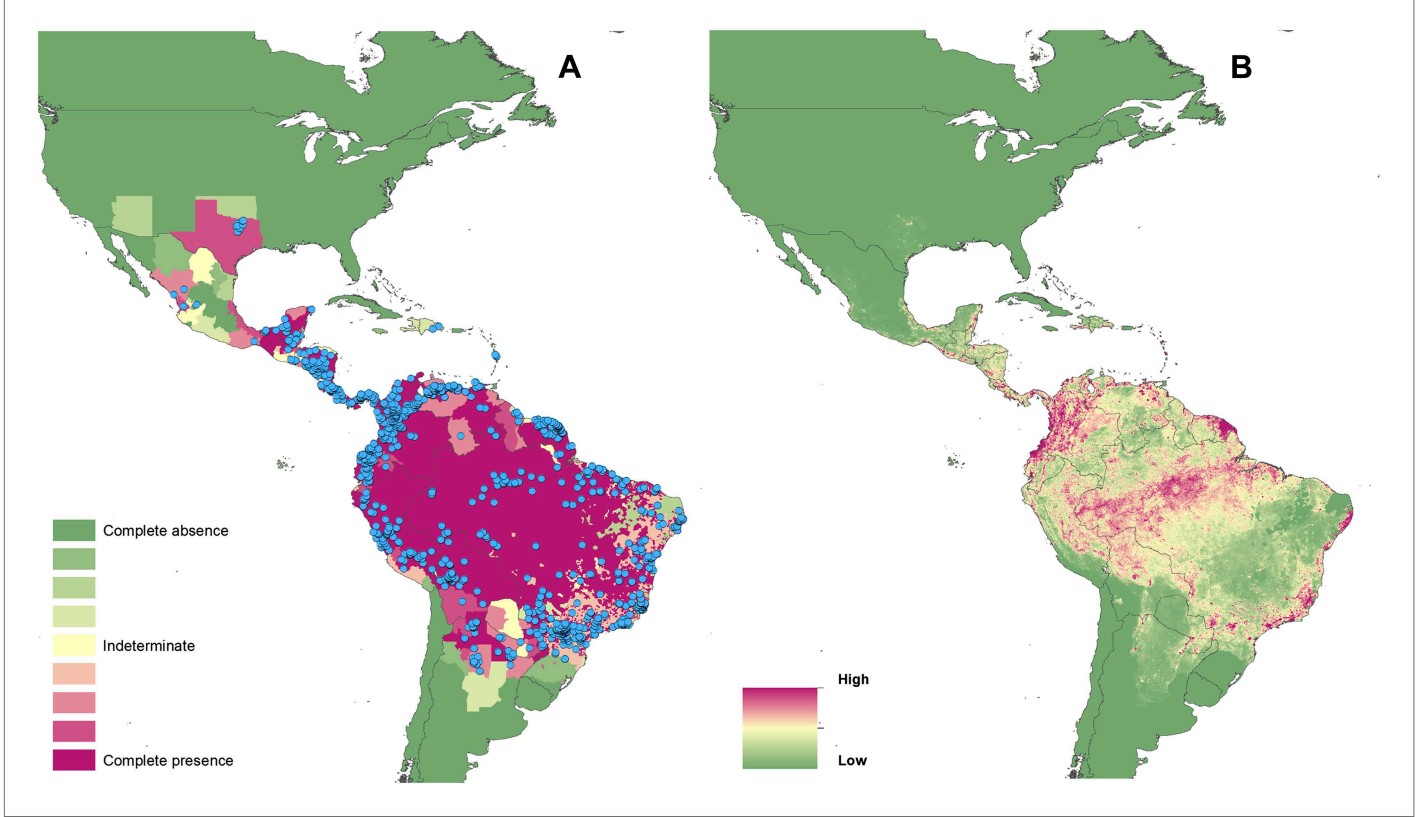

**Figure 1**. Reported and predicted distribution of cutaneous leishmaniasis in the New World. (**A**) Evidence consensus for presence of the disease ranging from green (complete consensus on the absence: −100%) to purple (complete consensus on the presence of disease: +100%). The blue spots indicate occurrence points or centroids of occurrences within small polygons. (**B**) Predicted risk of cutaneous leishmaniasis from green (low probability of presence) to purple (high probability of presence).

The following figure supplements are available for figure 1:

**Figure supplement 1**. Uncertainty associated with predictions in *Figure 1B*.

48.55% of VL distribution in the Old World and New World, respectively. Validation statistics for all models were high with a mean area under the receiver operator curve (AUC) above 0.97 and mean correlations above 0.85 for all models.

In the New World, CL is predicted to occur primarily within the Amazon basin and other areas of rainforest. By contrast, VL is predicted to occur mainly along the coastline of Brazil, with sporadic foci across the rest of Southern and Central America. Outside of their main foci, both diseases are strongly associated with urban and peri-urban areas, resulting in a focal distribution throughout much of the New World.

In the Old World, both CL and VL are predicted to be present from the Mediterranean Basin across the Near East to Northwest India, with a few foci in Central China as well as in a thin band of predicted risk across West Africa and in the Horn of Africa. The predicted distribution of VL also extends into Northeast India and China with a large predicted focus in the northwest.

The populations living in areas predicted to be subject to environmental risk of CL and VL are estimated to be 1.71 billion and 1.69 billion, respectively, approximately a quarter of the world's population. *Figure 4—figure supplement 4* compares these national estimates to the annual case incidence data from all countries for which at least one case *per annum* was estimated by *Alvar et al. (2012)*. There is a strong positive association between the two measures of disease occurrence. We provide estimates of the populations at risk in 90 countries for which no human cases of CL or VL were regularly reported (*Alvar et al., 2012*). A full table of this information is presented in the associated Dryad data set (doi: 10.5061/dryad.05f5h). For many of these countries, *Alvar et al. (2012)* reported a handful of

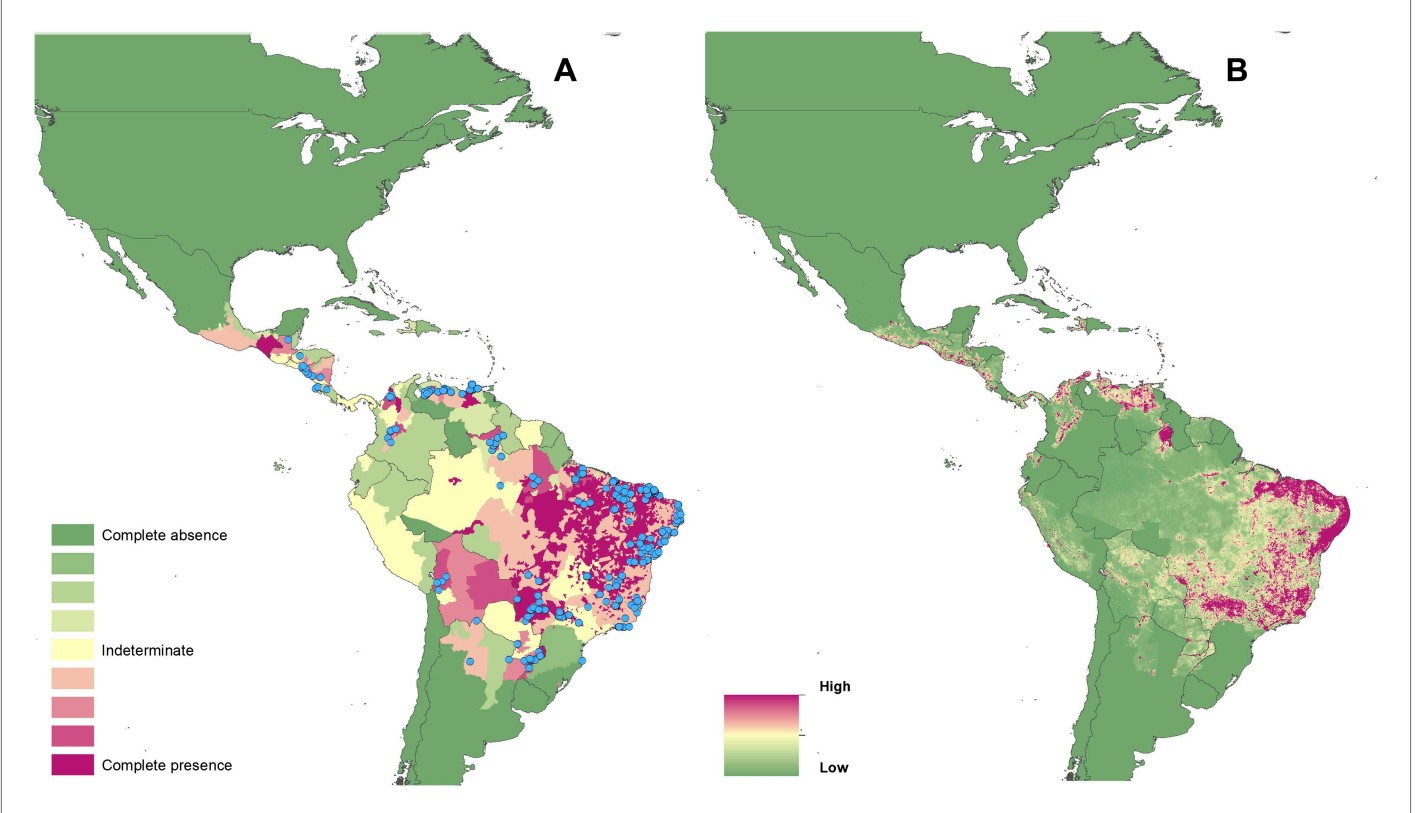

**Figure 2**. Reported and predicted distribution of visceral leishmaniasis in the New World. (**A**) Evidence consensus for presence of the disease ranging from green (complete consensus on the absence: −100%) to purple (complete consensus on the presence of disease: +100%). The blue spots indicate occurrence points or centroids of occurrences within small polygons. (**B**) Predicted risk of visceral leishmaniasis from green (low probability of presence) to purple (high probability of presence).

The following figure supplements are available for figure 2:

**Figure supplement 1**. Uncertainty associated with predictions in *Figure 2B*.

sporadic cases over the years indicating very rare occurrence of infection, whilst the remainder were countries with inconclusive evidence of disease presence or absence. It is important to note that the relationship between environmental risk and true incidence of disease remains to be elucidated; however the association between populations living in areas of environmental risk and national level estimates of incidence suggests that the modelled occurrence–incidence relationship approach used by *Bhatt et al. (2013)* for dengue could be applied if the necessary longitudinal cohort study data were available.

## Discussion

This work has compiled a large body of qualitative and quantitative information on the global distribution of the leishmaniases and employed a statistical modelling framework to generate the first published high-resolution global distribution maps of these diseases.

The evidence consensus maps provide a useful assessment of both global and regional knowledge of these diseases. Whilst in many countries consensus on presence or absence of the leishmaniases exists, in other areas, including large parts of Africa and many states in India, these assessments reveal significant uncertainty in assessing disease presence or absence using currently available evidence. It is in these data-poor countries that increased surveillance efforts should be concentrated to improve our knowledge of the global distribution of the leishmaniases. In some locations, cases have been reported as locally transmitted without the presence of proven vector species, which could indicate a false positive. However, the overall consensus score will reflect any uncertainty associated with the validity

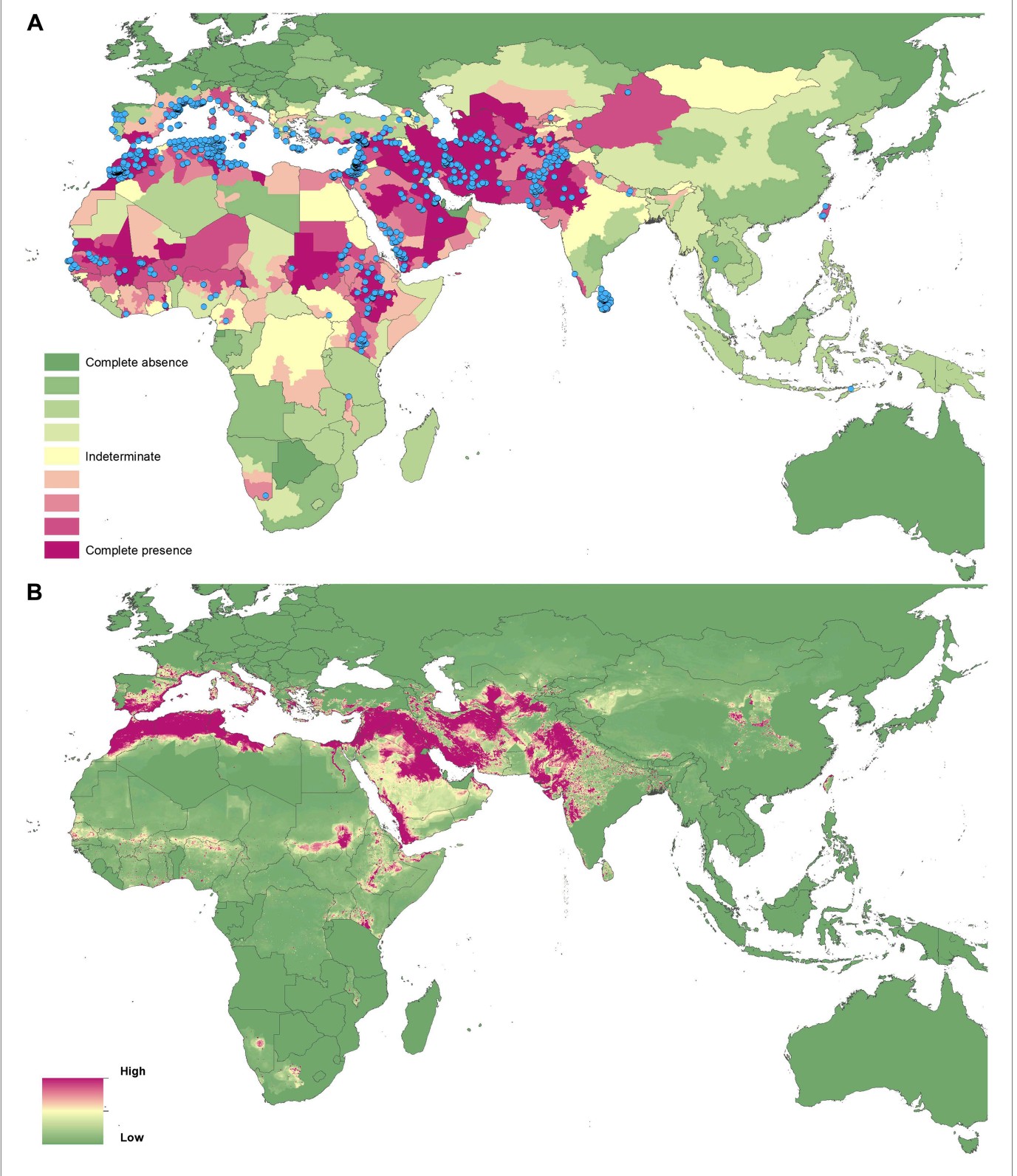

**Figure 3**. Reported and predicted distribution of cutaneous leishmaniasis in the Old World. (**A**) Evidence consensus for presence of the disease ranging from green (complete consensus on the absence: −100%) to purple (complete consensus on the presence of disease: +100%). The blue spots indicate

*Figure 3. Continued on next page*

*Figure 3. Continued*
occurrence points or centroids of occurrences within small polygons. (**B**) Predicted risk of cutaneous leishmaniasis from green (low probability of presence) to purple (high probability of presence).
The following figure supplements are available for figure 3:

**Figure supplement 1**. Uncertainty associated with predictions in *Figure 3B*.
**Figure supplement 2**. Reported and predicted distribution of cutaneous leishmaniasis in northeast Africa.
**Figure supplement 3**. Reported and predicted distribution of cutaneous leishmaniasis across the Near East, including Syria, Iran and Afghanistan.

of these reports; if multiple independent sources report autochthonous cases, this increased certainty will be reflected in a higher consensus score. Similarly, whilst the occurrence database contains data from across the globe, this data set is inevitably subject to spatial bias in reporting, with more data reported from more economically developed countries where we already have a good knowledge of the disease (e.g., Spain, France, and Italy).

The complexity and diversity of transmission cycles involving not just humans, but also a multitude of vectors and reservoirs, necessitated a modelling approach which can account for highly non-linear effects of covariates on probability of disease presence. The BRT modelling approach employed is able to do this and has previously been shown to produce highly accurate predictions across a wide range of species (*Elith et al., 2006*, *2008*). This ecological niche modelling approach is therefore able to deal with not only the variation in parasites causing infection, but also the various life-histories and habitat preferences associated with the different vector species.

A restriction of the BRT approach (in common with other species distribution modelling approaches) is the need for absence data in addition to occurrence data. Since reliable absence data were not available at this spatial scale, the incorporation of pseudo-data into the modelling framework was necessary. The methodology employed in this study attempted to minimise the problems this can cause, by using a probabilistic approach to generate the pseudo-data which incorporates the evidence consensus and distance from existing occurrence points. Similarly, reporting bias within the occurrence database is an issue with all presence-only species distribution models (*Peterson et al., 2011*). If bias is unaccounted for, there is the potential that the model merely reflects factors that correlate with the probability of reporting disease occurrence rather than the disease itself, such as healthcare expenditure (*Phillips et al., 2009*; *Syfert et al., 2013*). The pseudo-data selection procedures (which included information from both the occurrence data set and the less-biased evidence consensus map) coupled with the model ensembling approach aimed to minimise this potential source of bias.

The differences in the most important predictors of disease presence between the two forms of the disease and between the Old and New Worlds highlight the complex and spatially variable epidemiology of the leishmaniases. Similar to a recent study of the spatial predictors of dengue occurrence (*Bhatt et al., 2013*), environmental and socioeconomic factors were found to be important contributors to the distribution of both CL and VL. For VL, both Old World and New World distributions are driven by peri-urban (and to a lesser degree urban) land cover. This reflects recent trends observed, for instance, in Brazil and Bihar state in India, where areas of highest risk have been found in peridomestic settings (*Bern et al., 2010*; *Harhay et al., 2011*). This risk factor may well be linked back to aspects of vector bionomics, with many vectors in these regions associating with or near households in general (*Singh et al., 2008*; *Poche et al., 2011*; *Uranw et al., 2013*). Furthermore, whilst significant anthroponotic transmission of *L. donovani* occurs across parts of the Old World, zoonotic cycles of VL, primarily tied to canine hosts, dominate *L. infantum* transmission (*Chamaille et al., 2010*; *Ready, 2013*), with infection in dogs shown to be closely associated with human population density.

Important predictors of CL distribution differed markedly between the Old and New World. Whilst peri-urban land cover was the most important predictor of the disease in the Old World, in the New World temperature was the highest predictor, with abiotic factors predicting 74.18% of CL distribution. This difference in the relative importance of climatic drivers reflects the fact that in the Old World

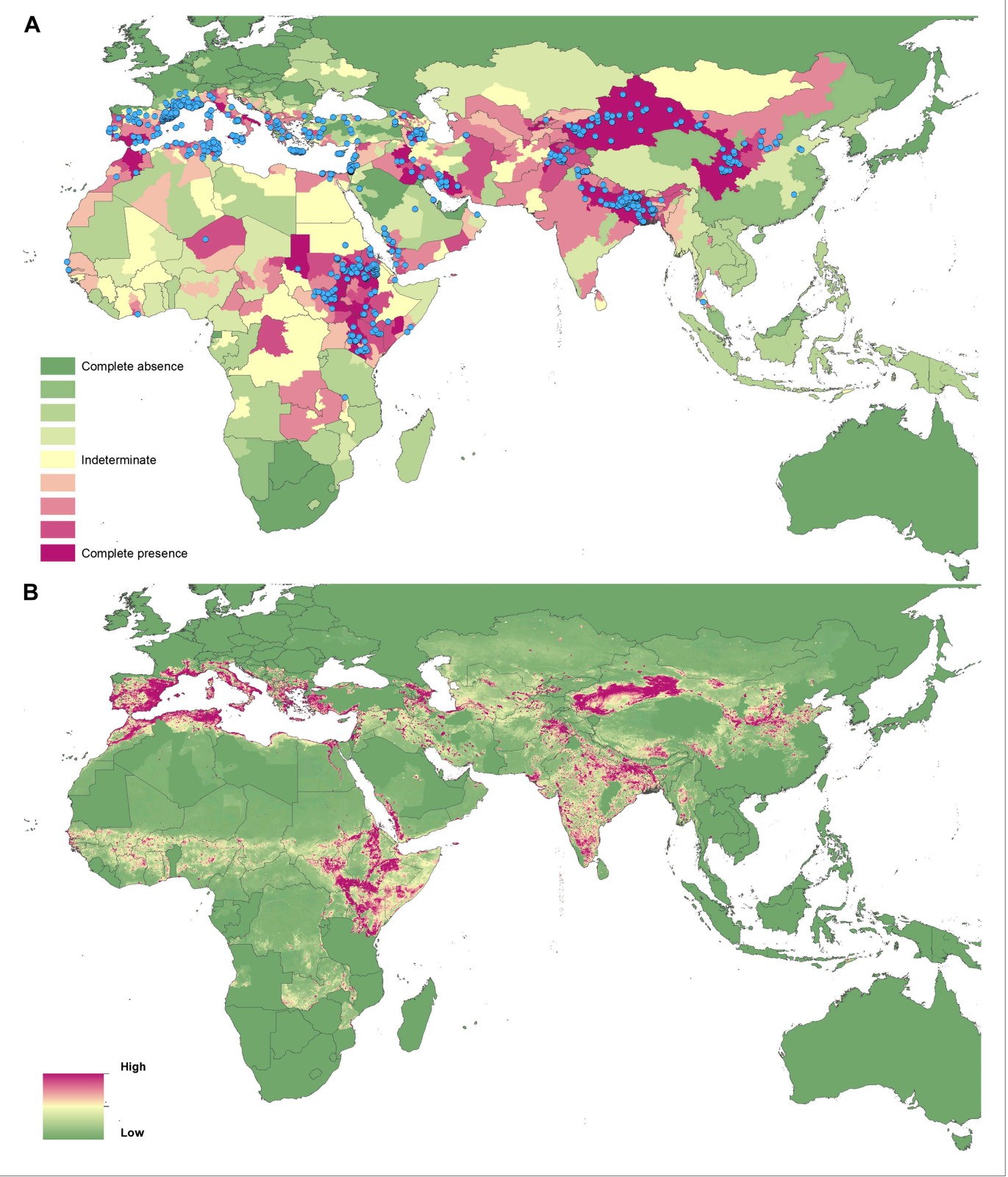

**Figure 4**. Reported and predicted distribution of visceral leishmaniasis in the Old World. (**A**) Evidence consensus for presence of the disease ranging from green (complete consensus on the absence: −100%) to purple (complete consensus on the presence of disease: +100%). The blue spots indicate occurrence points or centroids of occurrences within small polygons. (**B**) Predicted risk of visceral leishmaniasis from green (low probability of presence) to purple (high probability of presence).

*Figure 4. Continued on next page*

*Figure 4. Continued*

The following figure supplements are available for figure 4:

**Figure supplement 1**. Uncertainty associated with predictions in *Figure 4B*.

**Figure supplement 2**. Reported and predicted distribution of visceral leishmaniasis in northeast Africa.

**Figure supplement 3**. Reported and predicted distribution of visceral leishmaniasis in the Indian subcontinent.

**Figure supplement 4**. Population at risk estimates for leishmaniasis.

the main endemic CL areas are due to both anthroponotically transmitted *L. tropica* and zoonotic cycles of *L. major*, whereas in the New World the disease is primarily associated with sylvatic and zoonotic cycles with a variety of different *Leishmania* spp. and wild reservoir hosts implicated (*Ashford, 1996*; *Reithinger et al., 2007*; *WHO, 2010*; *Lima et al., 2013*; *Ready, 2013*).

The distribution maps represent a spatially refined assessment of the global environmental risk of leishmaniasis and provide a starting point for various public health activities including targeting areas for control and assessing disease burden. The maps compare favourably to the WHO Expert Committee on the Control of Leishmaniases outputs (*WHO, 2010*), have high model validation statistics and improve upon the existing body of work by providing a finer resolution of risk at a subnational level. Similarly, the countries indicated by *Alvar et al. (2012)* as having 90% of all VL and 75% of all CL cases, were all predicted by our maps to have risk for VL and CL, respectively.

There are a number of regions in which our maps do not correspond as closely to these previous findings. Regions such as Northwest China are predicted to have high risk for VL, though the low population densities in this area are likely to lead to very few cases and, given its remoteness, even fewer reported cases. Other regions, such as the Mediterranean coastline of Europe, are predicted to be highly suitable for leishmaniasis, but we see few human cases. This is because the maps presented predict the probability of disease presence in an area, rather than directly infer measures of incidence or burden, which can be influenced by a variety of other factors (e.g., in the Mediterranean coastline of Europe, VL has been associated with immunosuppression). The evidence consensus layer, used to mask out regions with high consensus on leishmaniasis absence, acts as a rough filter on the environmental risk

**Table 1.** Origin and spatial resolution of leishmaniasis occurrence data

**Origin and resolution of occurrence data**

| | Point data | Province level data | District level data | Total |
|---|---|---|---|---|
| *Cutaneous leishmaniasis* | | | | |
| Literature | 3680 | 879 | 1220 | 5779 |
| CNR-L | 531 | 47 | 31 | 609 |
| HealthMap | 31 | – | – | 31 |
| GenBank | 6 | – | 1 | 7 |
| Total | 4248 | 926 | 1252 | 6426 |
| *Visceral leishmaniasis* | | | | |
| Literature | 3050 | 1500 | 1068 | 5618 |
| CNR-L | 429 | 24 | 29 | 482 |
| HealthMap | 32 | 1 | – | 33 |
| GenBank | 3 | – | 1 | 4 |
| Total | 3514 | 1525 | 1098 | 6137 |

Each cell gives the number of occurrence records added to the data set by considering each additional datasource after removing duplicate records. Occurrence records are separated by spatial resolution—whether they are recorded as points (typically representing settlements) or as province level (admin 1) or district level (admin 2) data.

**Table 2.** Mean relative contribution of predictor variables to the ensemble BRT models of CL and VL in both the Old and New World

| Top predictors of CL | Relative contribution | Top predictors of VL | Relative contribution |
|---|---|---|---|
| *Old world* | | | |
| Peri-urban extents | 47.34 | Peri-urban extents | 51.50 |
| Minimum LST | 18.36 | Urban extents | 17.38 |
| Urban extents | 9.01 | Maximum NDVI | 7.87 |
| G-Econ | 7.33 | Minimum LST | 5.87 |
| Minimum Precipitation | 4.95 | Maximum Precipitation | 4.00 |
| *New World* | | | |
| Maximum LST | 36.91 | Peri-urban extents | 25.90 |
| Peri-urban extents | 18.61 | Urban extents | 21.24 |
| Maximum precipitation | 12.06 | Mean LST | 9.18 |
| Minimum precipitation | 6.21 | Mean NDVI | 7.83 |
| Minimum LST | 4.39 | Maximum LST | 6.40 |

LST = Land Surface Temperature, G-Econ = Geographically based Economic data, NDVI = Normalised Difference Vegetation Index.

maps. However, in order to model the true relationship between environmental risk and disease incidence, a global data set of geopositioned disease incidence data would be required; at present this is unavailable.

Estimates of the populations living in areas of environmental risk are therefore supplied as a proxy for the true burden of disease. However, they cannot be directly compared with other global estimates of the leishmaniases' disease burden, such as the WHO estimates of clinical burden of around 350 million (*WHO, 2010*). *Figure 4—figure supplement 4* shows a strong, positive relationship between population at risk estimates and estimated annual incidence from *Alvar et al. (2012)*. The exceptions to this relationship (e.g., Egypt, Nigeria, and Côte d'Ivoire) are all countries with indeterminate evidence consensus scores, indicating a genuine lack of knowledge regarding both the distribution and incidence of disease.

Previous estimates of the leishmaniases' global burden have been complicated by poor knowledge of the global distribution of the diseases (*Bern et al., 2008*; *Reithinger, 2008*). It is hoped that the maps presented here will help to increase the accuracy of future estimates. Ideally, future improvements to the global distribution maps presented here would distinguish between the different *Leishmania* species and sandfly vectors. Species-specific models at the same level of detail as those presented here are not currently possible due to a lack of suitable data. Developments in the use of 'big data' approaches to disease mapping (such as the incorporation of informal internet resources) may enable the construction of data sets which could be used in these analyses (*Hay et al., 2013*). A further complication with burden estimation is the epidemic nature of the disease, as evidenced by the national case time series in *Alvar et al. (2012)*, leading to significant interannual variation in burden. Therefore, any burden estimation would have to account for this and the temporal spread of data would therefore be critical.

It should be noted that non-environmental drivers of transmission and morbidity, such as HIV immunosuppression and risk of infection via blood transfusions and intravenous drug usage, are not incorporated into our present models. The maps presented here can help inform the wider discussion of these factors and their impact on leishmaniasis (e.g., by identifying regions with greater risk for HIV and leishmaniasis co-infection) (*Desjeux and Alvar, 2003*). Similarly, the niche based models used here could enable a decoupling of environmental from social factors to assess the importance of the latter on leishmaniasis transmission in particular areas. It may indeed be the case that in some specific localities it is these non-environmental risk factors that are the main determinants of disease distribution.

## Conclusions

These maps represent evidence-based estimates of the current global distribution of the leishmaniases incorporating a comprehensive occurrence database and a rigorous statistical modelling framework

with associated uncertainty statistics. We estimate that 1.71 billion and 1.69 billion individuals live in areas that are suitable for CL and VL transmission, respectively. These figures highlight the need for much greater awareness of this disease at a global scale. These maps provide an important baseline assessment and a strong foundation on which to base future burden estimates, target regions for control efforts and inform public health decisions.

## Materials and methods

A boosted regression tree (BRT) modelling framework was used to generate global predicted environmental risk maps for CL and VL. This framework required four key information components: (i) a map of the consensus of evidence for the global extents of the leishmaniases; (ii) a comprehensive data set of geopositioned CL and VL occurrence records; (iii) a suite of global, gridded data sets on environmental correlates of the leishmaniases; and (iv) pseudo-data to augment the occurrence records. In order to better capture the realised niche of these diseases, prediction by the model is restricted to those areas of known disease transmission, or where transmission is uncertain, as defined by the evidence consensus layer (i). The full procedures used to generate these components and the resulting risk and prevalence maps are outlined below.

### Evidence consensus

The methodology used for generating the definitive extents for the leishmaniases was adapted from work on dengue (*Brady et al., 2012*). Four primary evidence categories were used to determine a consensus on the presence or absence of the leishmaniases: (i) health reporting organisations; (ii) peer-reviewed evidence of local autochthonous transmission; (iii) case data; (iv) supplementary information. Cutaneous and visceral leishmaniasis were the two symptomatologies investigated: other forms of the disease were subset within these two – whilst VL contained cases of post-kala-azar dermal leishmaniasis, CL included diffuse, disseminated, and mucosal forms of the disease. Although limited amounts of data were available for some of these forms, their epidemiology is similar, and consequently this categorisation was seen as appropriate. Information was collected at provincial level (termed Admin 1 units by the Food and Agriculture Organization's (FAO) Global Administrative Unit Layers (GAUL) coding (*FAO, 2008*)) to better capture the focal nature of these diseases.

### Health Reporting Organisation Evidence (scores between −3 and +3)

Two health reporting organisations were referenced, the Global Infectious Diseases and Epidemiology Online Network (GIDEON) (*Edberg, 2005*) and the World Health Organization (WHO) (*WHO, 2010*). The status of disease was recorded for each Admin 1 unit as either present, absent or unspecified. If both reported the disease as present, +3 was scored, if both reported absence, −3 was scored, with +2/−2 scored if one reporting body did not specify the presence or absence of the disease. If the two disagreed, or both were non-specific, 0 was scored reflecting the lack of a consensus on the status of that region.

### Peer-reviewed evidence (scores between +2 and +6)

A review of reported leishmaniases' cases was performed. Using PubMed and Web of Knowledge with '[admin1 province] leish*' as the search parameters, articles from January 1960 until September 2012 were abstracted. Each abstract was imported into Endnote X4 and assessed for relevance. Papers that included reported cases on either CL or VL were then obtained. Cases were included if there was sufficient evidence to suggest that local autochthonous transmission had occurred. Where individuals from a non-endemic country had travelled to an endemic country (e.g., tourists and military personnel) and returned with an infection, this was included (as evidence for leishmaniasis in the foreign destination) since these typically represent immunologically naive individuals who have undergone more rigorous diagnostics in their home country, and thus represent a potentially more informed data source. Each paper was assessed for contemporariness and diagnostic accuracy. Contemporariness was graded in 3 bands: 2005–2012 = 3, 1997–2004 = 2 and 1997 and earlier = 1, as was diagnostic accuracy where 1 was scored for data that reported 'confirmed' cases without detailing methodologies implemented; 2 was scored where evidence of microscopy, serology, or the Montenegro skin test had been used; 3 was awarded to those studies that had used PCR or other molecular techniques (*Reithinger and Dujardin, 2007*). Contemporariness bins were based upon the potentially lengthy intrinsic incubation periods present with some *Leishmania* spp. as well as to accommodate the potential for epidemic cycles, where cases may only be detected in peak years and missed in the intervening

baseline periods. The most contemporary and diagnostically accurate papers were then subset to maximise the consensus score for any given area.

## Case data (scores between −6 and +6)

Case data were derived from reports on the leishmaniases provided by national health officials (*Alvar et al., 2012*). A threshold value of 12 CL cases and 7 VL cases in a given province in a given year was deemed suitable by the authors to distinguish significant disease events from sporadic cases within that region. If cases were reported at or above the threshold and were dated no later than 2005, +6 was scored. If data existed below this threshold, indicating sporadic cases, or data indicated a history of reported cases in the region but with no evidence of time period, scores were assigned stratified by total annual healthcare expenditure (HE) per capita at average US$ exchange rates (*WHO, 2011*). This was used as a proxy to determine genuine sporadic reporting from inadequate surveillance. Three categories were defined—HE Low (<$100), HE Medium ($100 ≤ HE < $500), and HE High (≥$500). If sporadic cases were reported in an HE Low country, +4 was scored, whilst in an HE Medium country, +2 was scored, and in an HE High country, 0 was scored. If there were no reported case data available, HE Low countries scored +2, HE Medium countries scored −2 and HE High countries scored −6 (*Brady et al., 2012*).

## Supplementary evidence

Supplementary evidence was provided in cases where a consensus on presence or absence could not be reached using the aforementioned evidence types, typically with areas where the consensus value was close to 0%. For these regions, additional literature searches were undertaken to determine whether known vector species or infected reservoir hosts were reported in the region. The justification for each provincial scenario is outlined in the associated online databases (Dryad data set doi: 10.5061/dryad.05f5h). In total, this assessment was required in 24 countries.

An overall consensus score for each administrative region was calculated by the sum of the scores in each category, divided by the maximum possible score, then expressed as a percentage. Consensus was defined as either complete (±75% to ±100%), good (±50% to ±74%), moderate (±25% to ±49%), poor (±1% to ±24%), or indeterminate (0%). Such a classification is intended more as a guide to the quality of evidence for the leishmaniases in an area, rather than as a strict classification of certainty. The full scores for each country are laid out in the associated online data sets (Dryad data set doi: 10.5061/dryad.05f5h).

## Brazil and Peru

The Brazilian Ministry of Health produces, via the Sistema de Informação de Agravos de Notificação (*SINAN, 2013*) reporting network, records of infections at the municipality level. This allowed for a more thorough evidence consensus to be performed at district level (termed Admin 2 *FAO, 2008*) within Brazil. As above, WHO and GIDEON status as well as peer-reviewed literature score were recorded, both aggregated to Admin 1 provincial level. Case data were then defined by the presence of a municipality reporting leishmaniasis between 2008 and 2011 inclusive, with positive reports scoring +6 and absence scoring −6. The overall consensus score was then calculated as above. In addition, provincial level case data for Peru was replaced by Ministry of Health information as it was more contemporary than that listed by *Alvar et al. (2012)*.

## Occurrence records

Two separate searches using PubMed and Web of Knowledge were undertaken using the search parameter "leish*," and including articles up to December 2012, and their respective abstracts, were filtered for relevance. From these searches, 4845 articles were collated, with data recorded at the resolution of either a point or Admin 1 or 2 polygon. These were then geo-positioned using Google Maps (https://maps.google.co.uk/). Each entry was evaluated to ensure that non-autochthonous cases and duplicate entries were eliminated. Each occurrence was assigned a start and end date based upon the content of the paper, used to define the time period over which occurrences were reported.

In addition to this resource, reports were taken from the HealthMap database (http://healthmap.org/en/). HealthMap is an online based infectious disease surveillance system that compiles data from informal data sources ranging from online news articles to ProMED reports (*Freifeld et al., 2008*). It parses information from these sources searching for relevant keywords, and then, using crowdsourcing and automated processes, geopositions those relating to the disease of interest. As of December 2012, a total of 690 leishmaniasis relevant articles were archived.

Searches were also performed on GenBank accessions, searching for archived genetic information from *Leishmania* spp. known to infect humans (*WHO, 2010*). If the host was identified as human, geographic indicators were assigned either as point, Admin 1 or Admin 2, based upon the information in the location tag. Tags at the national level were filtered out of the data set. In total, 563 accessions were associated with sub-national location details and added to the database.

Finally, data were provided from the curated strain archives of the Centre National de Référence des Leishmanioses (CNR-L) in Montpellier, France. In total, information about 3465 strains isolated from humans was provided, collected from between 1954 and 2013.

All data were geopositioned as precisely as possible, which resulted in both point-level data (referring to cities, towns or villages) as well as polygon-level data (provinces or districts) with area no greater than one square decimal degree. All data that had been manually geopositioned were checked to ensure coordinates were plausible and then occurrences were standardised annually to remove intra-annual duplicates, so that each individual record used in our model represented an occurrence of leishmaniasis infections in a given 5 km × 5 km location or administrative unit for one given year. As a result, the occurrence data were independent of burden; a location with 200 cases in one year has equal weighting in the model as a location with just one reported case, since it was only the presence of the disease being modelled.

## Environmental correlates

*Leishmania* spp. are known to have anthroponotic, zoonotic, or sylvatic transmission cycles in nature (*WHO, 2010*; *Ready, 2013*) which is apparent in the focal nature of the disease; however, there are some key features of the environment that are important in determining the distribution of disease across the globe. Numerous models have been constructed for local transmission scenarios implicating various environmental features from temperature and precipitation to socioeconomic factors relating to standards of living in villages in endemic foci. For the modelling process, a suite of global gridded environmental, biologically plausible, correlates was generated.

### Precipitation

Humidity and moisture, whether from rainfall or in the soil, have often been identified as important for the sandfly, with humidity influencing breeding and resting (*Ready, 2013*). Whilst relatively little is known about these breeding sites, of the few that have been identified, high humidity seems to be a common trait, including moist Amazonian soils, caves, animal burrows, and select human dwellings (*Killick-Kendrick, 1999*; *Feliciangeli, 2004*). Studies have indicated soil type and their moisture profiles as determinants of sandfly distribution (*Bhunia et al., 2010*; *Elnaiem, 2011*). Precipitation represents a good global proxy measure for moisture, and has been shown to play a prominent role in shaping disease distribution in previous leishmaniasis modelling efforts (*Thomson et al., 1999*; *Elnaiem et al., 2003*; *Bhunia et al., 2010*; *Chamaille et al., 2010*; *Gonzalez et al., 2010*, *2011*; *Elnaiem, 2011*; *Hartemink et al., 2011*; *Malaviya et al., 2011*).

Estimates of precipitation were obtained from the WorldClim database (www.worldclim.org). This resource, which is freely available online, provides data spanning from 1950 to 2000, describing monthly averages over this time, at a 1 km × 1 km resolution (*Hijmans et al., 2005*). Using this baseline, interpolated global climate surfaces were produced using ANUSPLIN-SPLINA software (*Hutchinson, 1995*). With the use of temporal Fourier analysis, seasonal and inter-annual variation in precipitation patterns, taken from the interpolated global surface, were used to calculate minimum and maximum monthly precipitation averages (*Rogers et al., 1996*; *Scharlemann et al., 2008*).

### Temperature

Temperature influences both the development of the infecting *Leishmania* parasite in the sandfly (*Hlavacova et al., 2013*) as well as the life cycle of the sandfly vectors. On one hand, studies have shown that with increasing temperatures, the metabolism of the sandfly increases, influencing oviposition, defecation, hatching, and adult emergence rates (*Kasap and Alten, 2005*; *Benkova and Volf, 2007*; *Guzman and Tesh, 2000*). On the other hand, higher temperatures have also been shown to increase mortality rates of adults (*Benkova and Volf, 2007*; *Guzman and Tesh, 2000*). Studies have integrated the effects of temperature on sandfly biting rates, sandfly mortality, and extrinsic incubation periods to produce maps of how the basic reproductive number of canine leishmaniasis varied spatially (*Hartemink et al., 2011*). Multiple studies have also implicated temperature (including maximum, minimum, and mean temperatures) as being an important explanatory variable for both sandfly and

disease distribution (**Thomson et al., 1999**; **Gebre-Michael et al., 2004**; **Bhunia et al., 2010**; **Chamaille et al., 2010**; **Fischer et al., 2010**; **Galvez et al., 2011**; **Fernandez et al., 2012**; **Branco et al., 2013**).

Using a similar methodology to generating precipitation surfaces, minimum, maximum, and mean monthly temperature values were generated (**Hijmans et al., 2005**).

## Normalised difference vegetation index (NDVI) and land cover

Vegetation provides many roles in sandfly habitat and survival, ranging from maintaining the necessary moisture profile for both immature stages and adults, to a sugar resource for both male and female sandflies (**Killick-Kendrick, 1999**; **Feliciangeli, 2004**; **Ready, 2013**). Moreover, vegetation is an important resource for many mammals that sandflies feed on, and that potentially are *Leishmania* reservoirs. The importance of considering NDVI was demonstrated with respect to the distribution of the reservoir *Psammomys obesus* (sand rat) and the distribution of its primary food, chenopods (**Toumi et al., 2012**). NDVI has been implicated as a key explanatory variable in the distribution of leishmaniasis cases in several studies (**Cross et al., 1996**; **Thomson et al., 1999**; **Elnaiem et al., 2003**; **Gebre-Michael et al., 2004**; **Elnaiem, 2011**; **Hartemink et al., 2011**; **Bhunia et al., 2012**; **Toumi et al., 2012**; **de Oliveira et al., 2012**).

The Advanced Very High Resolution Radiometer (AVHRR) NDVI product uses the spectral reflectance of AVHRR channels 1 and 2 (visible red and near infrared wavelength) to quantitatively assess the level of photosynthesising vegetation in a region (**Hay et al., 2006**). Using this data, compiled over multiple time intervals, patterns of NDVI were extracted for each gridded 1 km × 1 km cell.

## Poverty

Neglected tropical diseases and poverty are often found to be linked and the use of a purely economic variable was chosen to act as a proxy for a variety of important global risk factors for disease, including malnutrition, housing quality, and living with domesticated animals (**Bern et al., 2010**; **Boelaert et al., 2009**; **Herrero et al., 2009**; **Malafaia, 2009**; **Zeilhofer et al., 2008**).

The G-Econ database (gecon.yale.edu) takes economic data, at the smallest administrative division available, and spatially rescales these data to create a 1° × 1° gridded surface of the globe (**Nordhaus, 2006**, **2008**). This rescaling estimates the gross cell product of each grid cell, conceptually similar to gross domestic product, referring to the total market value of all final goods and services produced within 1 year, and can be considered as an indicator of overall standard of living within that area. Some cells provided multiple data; in these scenarios the best-quality information, as outlined by the quality field associated with the data, was used to select one value. All gross cell product values were then adjusted using purchasing power parity in US$ for the years 1990, 1995, 2000, and 2005, using national aggregates estimated by the World Bank (**Nordhaus, 2006**) and computed the mean across all years for each gridded cell globally. This adjusted measure was used as the indicator of poverty in the model.

## Urbanisation

Over the last few decades, there has been a tendency for the leishmaniases having a sylvatic/zoonotic transmission cycle to transition into the urban and peri-urban environment in response to increasing urbanisation trends (**Harhay et al., 2011**). The increasing overlap in habitat between suitable human and animal hosts and multiple available resting sites for adults can allow for transmission of disease to occur relatively easily (**Singh et al., 2008**; **Poche et al., 2011**; **Uranw et al., 2013**).

The Gridded Population of the World version 3 (GPW3) population density database projected for 2010 was used. The core Global Rural–Urban Mapping Project Urban Extents surface used night-time light satellite imagery to differentiate urban areas (**Balk et al., 2006**); GPW3 is a revision which updates the criteria for urban areas to those areas where population density is greater than or equal to 1000 people per km². Using the most up-to-date national censuses available and other demographic data resolved to the smallest available administrative unit, a gridded surface of 5 km × 5 km cells was generated. Each pixel could then be classified as urban, peri-urban, or rural.

## Modelling with boosted regression trees

The boosted regression trees (BRT) methodology employed for mapping the leishmaniases is a variant of the model used in a previous analysis of dengue (**Bhatt et al., 2013**). Boosted regression tree modelling combines both regression trees, which build a set of decision rules on the predictor variables by portioning the data into successively smaller groups with binary splits (**De'ath, 2007**;

*Elith et al., 2008*), and boosting, which selects the tree that minimises the loss of function, to best capture the variables that define the distribution of the input data. The core BRT setup followed standard protocol already defined elsewhere (*Elith et al., 2008*; *Bhatt et al., 2013*).

## Pseudo-data generation

As BRT requires both the presence and absence data, the latter which is often hard to collate in an unbiased manner, pseudo-data had to be generated (*Elith et al., 2008*). There is no general consensus on how best to generate pseudo-data (*Bhatt et al., 2013*); however, several factors of the generation process are known to influence the predicted distribution and thus can be sources of potential bias (*Phillips et al., 2009*; *Van Der Wal et al., 2009*; *Phillips and Elith, 2011*; *Barbet-Massin et al., 2012*). In order to minimise such effects, pseudo-absence selection was directly related to the evidence consensus layer and restricted to a maximum distance ($\mu$) from any occurrence point. Pseudo-presence data was also incorporated, again informed by the evidence consensus layer, to compensate for poor surveillance capacity in low prevalence regions. As in *Bhatt et al. (2013)* points were randomly located in regions above an evidence consensus threshold of −25, with regional placement probability weighted by evidence consensus scores, so that regions with higher evidence consensus contained more pseudo-presences than lower scoring areas. Since the occurrence data set is from a wide range of sources and institutions, this procedure aims to mitigate sampling bias. By referencing the evidence consensus layer for pseudo-data selection, detection bias was also mitigated.

## 'Ensemble' analysis

There is no definitive procedure for choosing the best number of pseudo-data points to generate the most accurate predictive map. To account for the impact that these parameters might have on the model predictions, an ensemble BRT model was constructed with multiple BRT submodels fitted using pseudo-data points generated using different combinations of parameters $n_a$, $n_p$, and $\mu$. The numbers of pseudo-absences ($n_a$) and pseudo-presences ($n_p$) were defined as a proportion of the total number of actual data occurrence records (6426 and 6137 for CL and VL). The proportions used for generating pseudo-absences were 2:1, 4:1, 6:1, 8:1, and 10:1, and pseudo-presences were 0.025:1, 0.05:1 and 0.1:1. The pseudo-data were also generated within a restricted maximum distance ($\mu$) from any actual presence point, and $\mu$ was varied through 5 distances: 5, 10, 15, 20, and 25 arc degrees. All combinations of these parameter values resulted in a total of 75 ($5n_a \times 3n_p \times 5\mu$) individual input data sets and BRT submodels (making up the BRT ensemble).

For each disease, the 75 BRT submodels were used to predict a range of different risk maps (each at 5 km × 5 km resolution), and these were combined to produce a single mean ensemble risk map for each disease, also allowing for computation of the associated range of uncertainty in these predictions for every 5 km × 5 km pixel as shown in *Figure 1—figure supplement 1*, *Figure 2—figure supplement 1*, *Figure 3—figure supplement 1*, *Figure 4—figure supplement 1*. For both diseases, the New World (the Americas) and Old World (Eurasia and Africa) were modelled separately in order to account for and explore any differences in the epidemiology of the diseases between these regions. This was done to differentiate the potential effect that the different vectors namely *Lutzomyia* spp. in the New World and *Phlebotomus* spp. in the Old World and their varying life histories, might have on the distribution of the diseases within these regions.

## Summarising the BRT model

The relative importance of predictor variables was quantified for the final BRT ensemble. Relative importance is defined as the number of times a variable is selected for splitting, weighted by the squared improvement to the model as a result of each split and averaged over all trees (*Friedman, 2001*). These contributions are scaled to sum to 100, with a higher number indicating a greater effect on the response. To evaluate the ensemble's predictive performance, we used the area under the receiver operator curve (AUC) (*Fleiss et al., 2003*)—the area under a plot of the true positive rate versus false positive rate, reflecting the ability to discriminate between the presence and absence. An AUC value of 0.5 indicates no discriminative ability, and a value of 1 indicates perfect discrimination.

It is important to note that this distribution modelling technique assesses pixel level risk, rather than population level risk. As such, the ensemble evaluates the likelihood of leishmaniasis presence based upon the covariates supplied. In reality, some other factors, such as national healthcare provisioning and standards of living will influence the true observed burden. Therefore, whilst these two levels of

risk are inherently related, additional information, namely incidence data from many different populations, is required in order to assess the link quantitatively (*Bhatt et al., 2013*).

## Estimation of population living in areas of environmental risk

Population living in areas of risk was estimated by using a threshold probability to reclassify the probabilistic risk maps into a binary risk map, then extracting the total human population in the 'at risk' areas using a gridded data set of human population density from 2010 (*Balk et al., 2006*; *CIESIN/IFPRI/WB/CIAT, 2007*). The threshold value was set such that 95% of the point occurrence records fell within the at risk area. 5% of occurrence points were allowed to fall outside the predicted risk area to account for errors which could have arisen either from errors in the occurrence data set or from inaccuracies in the predicted risk maps.

For external validation, this population at risk information was compared to national reported annual cases (*Alvar et al., 2012*) to produce *Figure 4—figure supplement 4*. In these figures, the points represent the mean value of the estimated annual incidence reported taking into account the authors estimates of underreporting rates (*Alvar et al., 2012*). The upper and lower limits to these estimates are reflected by the bars around each point. Note that these figures use a log-scale on each axis and that only countries with non-zero estimates by *Alvar et al. (2012)* are included.

The threshold probabilities of occurrence used to define 'at risk' were as follows: NW CL—0.22, OW CL—0.19, NW VL—0.42, OW VL—0.19.

## Acknowledgements

DMP is funded by a Sir Richard Southwood Graduate Scholarship from the Department of Zoology at the University of Oxford. SIH is funded by a Senior Research Fellowship from the Wellcome Trust (#095066) which also supports KAD and KEB. NG is funded by a grant from the Bill & Melinda Gates Foundation (#OPP1053338). PWG is a Medical Research Council (UK) Career Development Fellow (#K00669X) and receives support from the Bill and Melinda Gates Foundation (#OPP1068048) which also supports SB. OJB is funded by a BBSRC studentship. JPM is funded by, and SIH and OJB acknowledge the support of, the International Research Consortium on Dengue Risk Assessment Management and Surveillance (IDAMS, European Commission 7[th] Framework Programme (#21803) http://www.idams.eu). JSB, CF and SRM acknowledge funding from NIH National Library of Medicine (#R01LM010812). The funders had no role in study design, data collection and analysis, decision to publish, or preparation of the manuscript.

## Additional information

### Funding

| Funder | Grant reference number | Author |
| --- | --- | --- |
| University Of Oxford | Department of Zoology, Sir Richard Southwood Graduate Scholarship | David M Pigott |
| Wellcome Trust | 095066 | Kirsten A Duda, Katherine E Battle, Simon I Hay |
| Bill and Melinda Gates Foundation | #OPP1053338 | Nick Golding |
| Biotechnology and Biological Sciences Research Council | Studentship | Oliver J Brady |
| European Commission | #21803 | Oliver J Brady, Jane P Messina, Simon I Hay |
| National Institutes of Health | R01LM010812 | John S Brownstein, Clark C Freifeld, Sumiko R Mekaru |
| Bill and Melinda Gates Foundation | #OPP1068048 | Samir Bhatt, Peter W Gething |

| Funder | Grant reference number | Author |
|---|---|---|
| Medical Research Council | #K00669X | Peter W Gething |

The funders had no role in study design, data collection and interpretation, or the decision to submit the work for publication.

## Author contributions

DMP, SIH, Conception and design, Acquisition of data, Analysis and interpretation of data, Drafting or revising the article; SB, NG, PWG, DBG, Conception and design, Analysis and interpretation of data, Drafting or revising the article; KAD, KEB, YB, FP, JSB, CF, SRM, MFM, Acquisition of data, Drafting or revising the article; OJB, PB, Acquisition of data, Analysis and interpretation of data, Drafting or revising the article; JPM, RR, Analysis and interpretation of data, Drafting or revising the article

## Author ORCIDs

David M Pigott, http://orcid.org/0000-0002-6731-4034
Simon I Hay, http://orcid.org/0000-0002-0611-7272

## Additional files

### Major dataset

The following dataset was generated:

| Author(s) | Year | Dataset title | Dataset ID and/or URL | Database, license, and accessibility information |
|---|---|---|---|---|
| Pigott DM, Bhatt S, Golding N, Duda KA, Battle KE, Brady OJ, Messina JP, Balard Y, Bastien P, Pratlong F, Brownstein JS, Freifeld C, Mekaru SR, Gething PW, George DB, Myers MF, Reithinger R | 2014 | Data from: Global Distribution Maps of the Leishmaniases | 10.5061/dryad.05f5h | Available at Dryad Digital Repository under a CC0 Public Domain Dedication. |

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
