## [Decision Letter]

Thank you for sending your work entitled “Global Distribution Maps of the
Leishmaniases” for consideration at *eLife*. Your article has
been favorably evaluated by Prabhat Jha (Senior editor), a Reviewing editor, and 3
reviewers.

The Reviewing editor and the other reviewers discussed their comments before we
reached this decision, and the Reviewing editor has assembled the following comments
to help you prepare a revised submission.

1) This work on mapping of leishmaniasis is unique and impressive in its scope and
depth, and makes for the most comprehensive overview of the leishmaniasis burden
worldwide to date. Rightly so, climatic as well as socio-economic factors were taken
into account to predict the risk of leishmaniasis. Indeed, this work will be able to
guide health authorities in future surveillance activities. However, to serve this
purpose it would be extremely helpful if the maps were presented in a format where
it would be possible to 'zoom in' so that the geographical locations can
be more easily identified. Specifically, the detail of the global prediction maps in
Figures 3 and 4 are difficult to
see. The authors could consider including larger insert maps for the major endemic
areas, e.g. east Africa and Indian subcontinent for VL.

2) In Asia as well as in Africa, VL caused by *L.donovani* typically
presents as epidemics, with the case load rising and falling over a period of 5-10
years, probably dependent on climatic factors as rainfall, and thus presenting as a
varying burden to countries. Similarly, CL caused by *L. major* and
*L. tropica* are prone to epidemics. Please address in the
Discussion.

3) A complete data review was used for establishing the evidence consensus for
presence of leishmaniasis. However, in any country where the appropriate vector for
transmission has not been confirmed according to the criteria set in 'Control
of the Leishmaniasis' (WHO, TRS 949, 2010) it cannot be assumed either that
leishmaniasis is endemic, or that the area is suitable for leishmaniasis
transmission. It is unclear whether this has been taken into account; if not, please
refer to 'Control of the Leishmaniasis' where expert consensus on vector
presence in each country is compiled. An example is Taiwan: according to map 3A
there is an area of confirmed CL presence, yet the vector for transmitting
*L. tropica* has not been confirmed.

Minor comments:

4) Differences in sandfly ecology. Different sandfly species have distinct ecologies
and habitat preference (for example *Phlebotomus orientalis* and
*P. martini* in east Africa) and the authors should explain how
such differences are taken into account.

5) Classification of contemporariness. Provide a justification as to the year bins
used.

6) Pseudo-presence data. The generation of such data was not clear and the authors
should provide further details.

7) “We provide estimates of the populations at risk in 90 countries for which
no human cases of CL or VL were reported.” This is interesting information
but we did not find it presented obviously in the article.

8) Furthermore, significant anthroponotic transmission of both *L.
infantum* and *L. donovani* occurs across much of the Old
World with zoonotic cycles of VL primarily tied to canine hosts. While transmission
of *L. donovani* is anthroponotic, there is no anthroponotic
transmission of *L. infantum* where transmission is entirely zoonotic
via canine hosts.

9) “In the Old World the main endemic CL areas are due to
anthroponotically-transmitted *L. tropica*”. True, but a
significant case load is also caused by zoonotic *L. major* in Old
World CL.

---

## [Author Response]

*1) This work on mapping of leishmaniasis is unique and impressive in its
scope and depth, and makes for the most comprehensive overview of the
leishmaniasis burden worldwide to date. Rightly so, climatic as well as
socio-economic factors were taken into account to predict the risk of
leishmaniasis. Indeed, this work will be able to guide health authorities in
future surveillance activities. However, to serve this purpose it would be
extremely helpful if the maps were presented in a format where it would be
possible to 'zoom in' so that the geographical locations can be more
easily identified. Specifically, the detail of the global prediction maps
in*
Figures 3 and 4
*are difficult to see. The authors could consider including larger insert
maps for the major endemic areas, e.g. east Africa and Indian subcontinent for
VL*.

We agree with the reviewers and have therefore supplied 4 figure supplements
detailing a more close up view of CL in northeast Africa and across the Near East,
and of VL in northeast Africa and the Indian subcontinent. These panels depict areas
of the highest incidence of both CL and VL. In addition, maps can be provided to
individuals upon request to the authors.

*2) In Asia as well as in Africa, VL caused by* L. donovani
*typically presents as epidemics, with the case load rising and falling
over a period of 5-10 years, probably dependent on climatic factors as rainfall,
and thus presenting as a varying burden to countries. Similarly, CL caused
by* L. major *and* L. tropica *are prone to
epidemics. Please address in the Discussion*.

Whilst this is a crucial component of burden estimation, we believe this
doesn’t directly impact on the BRT models used, since these are reliant on
presence/absence data, not total numbers of cases in a given year. In parts of the
evidence consensus generation process where temporal data is important, such as
contemporariness score for peer-review data and the scoring of case data series, the
temporal divisions used to analyse these data is sufficiently broad to accommodate
most inter-annual variation. However, we have added a section in the Discussion
highlighting that this characteristic will further complicate potential burden
estimation:

“A further complication with burden estimation is the epidemic nature of the
disease, as evidenced by the national case time series in [1]*,* leading
to significant interannual variation in burden. Therefore, any burden estimation
would have to account for this and the temporal spread of data would therefore be
critical.”

*3) A complete data review was used for establishing the evidence consensus
for presence of leishmaniasis. However, in any country where the appropriate
vector for transmission has not been confirmed according to the criteria set in
'Control of the Leishmaniasis' (WHO, TRS 949, 2010) it cannot be
assumed either that leishmaniasis is endemic, or that the area is suitable for
leishmaniasis transmission. It is unclear whether this has been taken into
account; if not, please refer to 'Control of the Leishmaniasis' where
expert consensus on vector presence in each country is compiled. An example is
Taiwan: according to map 3A there is an area of confirmed CL presence, yet the
vector for transmitting* L. tropica *has not been
confirmed*.

From the outset we set out to model reported cases of leishmaniasis infection and
disease in humans, therefore the evidence consensus was primarily driven by evidence
of local autochthonous transmission of the disease. Whilst in some cases the vector
species is unknown or unproven, this may just as equally reflect the rarity of the
disease in this area (and hence little knowledge available on vector species) rather
than necessarily the suggestion of local transmission being incorrect. As a result,
we chose to prioritise evidence of autochthonous cases of disease. Where there was
insufficient evidence pertaining to human cases, information concerning vector and
reservoir distributions was also considered, and this was taken from reports in the
literature. In the regions where this was considered, the findings were consistent
with the WHO Technical Report, apart from the presence of sandflies (not proven to
transmit disease locally) in two regions of Tanzania. In the specific case of
Taiwan, several cases have been reported as being locally derived (as outlined in
the evidence consensus tables in the Dryad dataset) and therefore the evidence
consensus scores Taiwan as likely to have the disease present. The region scores
+53.33%, therefore indicating that this is not unanimously agreed upon by the
various sources consulted, however this is supported by GIDEON and the [1] paper.
The WHO technical report also indicates that *P. kiangsuensis* could
act as a potential vector. In order to better clarify this situation, we have
clarified the text in three places:

a) “(ii) peer-reviewed evidence of local autochthonous
transmission”

b) “Cases were included if there was sufficient evidence to suggest that local
autochthonous transmission had occurred”

c) “In some locations, cases have been reported as locally transmitted without
the presence of proven vector species, which could indicate a false positive.
However, the overall consensus score will reflect any uncertainty associated with
the validity of these reports; if multiple independent sources report autochthonous
cases, this increased certainty will be reflected in a higher consensus
score.”

Minor comments:

*4) Differences in sandfly ecology. Different sandfly species have distinct
ecologies and habitat preference (for example* Phlebotomus orientalis
*and* P. martini *in east Africa) and the authors should
explain how such differences are taken into account.*

We have reinforced the relevant section in the Discussion relating to the flexibility
of BRT and how it can deal with complexity:

“The complexity and diversity of transmission cycles involving not just
humans, but also a multitude of vectors and reservoirs, necessitated a modelling
approach which can account for highly non-linear effects of covariates on
probability of disease presence. The BRT modelling approach employed is able to do
this and has previously been shown to produce highly accurate predictions across a
wide range of species. This ecological niche modelling approach is therefore able to
deal with not only the variation in parasites causing infection, but also the
various life-histories and habitat preferences associated with the different vector
species.”

*5) Classification of contemporariness. Provide a justification as to the year
bins used*.

We have added the following in light of this comment:

“Contemporariness bins were based upon the potentially lengthy intrinsic
incubation periods present with some *Leishmania* spp. as well as to
accommodate the potential for epidemic cycles, where cases may only be detected in
peak years and missed in the intervening baseline periods.”

*6) Pseudo-presence data. The generation of such data was not clear and the
authors should provide further details*.

We have added some more details to those already listed in the document to help
clarify. The manner in which the pseudo-presence data was incorporated as a
numerical parameter in the BRT process can be found in the paragraph entitled
Ensemble analysis:

“As in [9] points were randomly located in regions above an evidence
consensus threshold of -25, with regional placement probability weighted by evidence
consensus scores, so that regions with higher evidence consensus contained more
pseudo-presences than lower scoring areas.”

*7) “We provide estimates of the populations at risk in 90 countries
for which no human cases of CL or VL were reported.” This is interesting
information but we did not find it presented obviously in the
article*.

We have provided via the Dryad dataset associated with this output, tables detailing
national estimates of populations living in areas of environmental risk of
leishmaniasis. We have also inserted the following section as a synoptic overview of
these countries:

“A full table of this information is presented in the associated Dryad
database (doi:10.5061/dryad.05f5h). For many of these countries, [1] reported a handful of
sporadic cases over the years indicating very rare occurrence of infection, whilst
the remainder were countries with inconclusive evidence of disease presence or
absence.”

*8) Furthermore, significant anthroponotic transmission of both* L.
infantum *and* L. donovani *occurs across much of the Old
World with zoonotic cycles of VL primarily tied to canine hosts. While
transmission of* L. donovani *is anthroponotic, there is no
anthroponotic transmission of* L. infantum *where transmission is
entirely zoonotic via canine hosts*.

We have changed this section to reflect this comment:

“Furthermore, whilst significant anthroponotic transmission of *L.
donovani* occurs across parts of the Old World, zoonotic cycles of VL,
primarily tied to canine hosts, dominate *L. infantum* transmission
(16; 65), with infection in dogs shown to
be closely associated with human population density.”

*9) “In the Old World the main endemic CL areas are due to
anthroponotically-transmitted L. tropica”. True, but a significant case
load is also caused by zoonotic* L. major *in Old World
CL*.

We have changed this section to reflect the fact that climatic factors have differing
relative influences between the Old World and New World. Table 2 demonstrates that whilst periurban extents are the
most important predictor of Old World CL, temperature and to a lesser extent,
precipitation, have a non-negligible influence, reflecting the two core
epidemiologies present with *L. tropica* and *L.
major*:

“This difference in the relative importance of climatic drivers reflects the
fact that in the Old World the main endemic CL areas are due to both
anthroponotically-transmitted *L. tropica* as well as zoonotic cycles
of *L. major*, whereas in the New World the disease is primarily
associated with sylvatic and zoonotic cycles with a variety of different
*Leishmania* spp. and wild reservoir hosts implicated (2; 51; 65; 68;
82).”